# SCICO: Hierarchical Cross-Document Coreference for Scientific Concepts

**Arie Cattan**[1][*]                                         ARIE.CATTAN@GMAIL.COM
**Sophie Johnson**[2]                                         SOPHIEJ@ALLENAI.ORG
**Daniel Weld**[2,3]                                          DANW@ALLENAI.ORG
**Ido Dagan**[1]                                              DAGAN@CS.BIU.AC.IL
**Iz Beltagy**[2]                                             BELTAGY@ALLENAI.ORG
**Doug Downey**[2]                                            DOUGD@ALLENAI.ORG
**Tom Hope**[2,3]                                             TOMH@ALLENAI.ORG

[1]*Computer Science Department, Bar Ilan University, Ramat-Gan, Israel*

[2]*Allen Institute for Artificial Intelligence*

[3]*Paul G. Allen School for Computer Science & Engineering, University of Washington*

[*]*Work done during an internship at AI2.*

## Abstract

Determining coreference of concept mentions across multiple documents is a fundamental task in natural language understanding. Previous work on cross-document coreference resolution (CDCR) typically considers mentions of events in the news, which seldom involve abstract technical concepts that are prevalent in science and technology. These complex concepts take diverse or ambiguous forms and have many hierarchical levels of granularity (e.g., tasks and subtasks), posing challenges for CDCR. We present a new task of *Hierarchical* CDCR (H-CDCR) with the goal of *jointly* inferring coreference clusters and hierarchy between them. We create SCICO, an expert-annotated dataset for H-CDCR in scientific papers, 3X larger than the prominent ECB+ resource. We study strong baseline models that we customize for H-CDCR, and highlight challenges for future work.

## 1. Introduction

Cross-document coreference resolution (CDCR) identifies and links textual mentions that refer to the same entity or event across multiple documents. This fundamental task has seen much work recently [Choubey and Huang, 2017, Kenyon-Dean et al., 2018, Barhom et al., 2019, Cattan et al., 2021a,b, Caciularu et al., 2021] and can benefit various downstream applications such as multi-hop question answering [Dhingra et al., 2018, Wang et al., 2019], multi-document summarization [Falke et al., 2017], and discovery of cross-document relations [Hope et al., 2017, 2020, 2021].

Existing datasets for CDCR, such as ECB+ [Cybulska and Vossen, 2014], focus on mentions of news events involving concrete entities such as people or places. Abstract technical concepts are largely unexplored despite their prevalence in domains such as science and technology, and can pose significant challenges for CDCR: they often take diverse forms (e.g., *class-conditional image synthesis* and *categorical image generation*) or are ambiguous (e.g., *network architecture* in AI vs. systems research). These complex concepts also have many hierarchical levels of granularity, such as tasks that can be divided into finer-grained subtasks, where reference to a specific concept entails also a reference to the higher-level concept (e.g., *CRF* entails the *sequence tagging* task), unlike events and entities in ECB+ that are treated as non-hierarchical.

In this paper, we formulate a novel task of *hierarchical CDCR* (H-CDCR). The task is to infer (1) *cross-document entity coreference* clusters of concept mentions in scientific papers, and (2) *referential hierarchy between clusters*, where referring to a child cluster entails reference to the parent. Figure 1 shows the structure we aim to induce given mentions in context. Our task is the first to consider *unified* CDCR and hierarchy between *clusters* of mentions in context, unlike most work focusing on lexicon-level taxonomies over uncontextualized *words* [Shwartz et al., 2016, Zhang et al., 2018].

H-CDCR in science can support many applications. One example illustrating our setting is faceted query by example [Mysore et al., 2021], where given a paper, a user may highlight a specific span (facet) (e.g., *network architecture*), and explore other papers that refer to the same concept while resolving ambiguity and clustering diverse surface forms. Selecting a facet may also show concept hierarchies enabling users to browse and navigate collections [Hope et al., 2020] (e.g., by viewing a list of many types of network architectures, automatically identified). See §6 for more discussion of related areas and problems that could benefit from H-CDCR in the scientific domain.

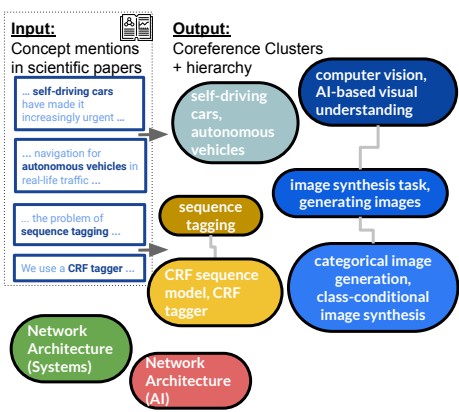

Figure 1: Given a set of mentions of scientific concepts, our goal is to induce clusters of coreferring mentions and a referential hierarchy between clusters indicating that reference to a child concept (e.g., *CRF model*) entails reference to a parent concept (*sequence tagging*).

To advance research in this area and enable supervised model training and evaluation, we create a new large-scale dataset named SCICO (**S**cientific **C**oncept **I**nduction **C**orpus), which is annotated by domain experts. SCICO consists of clusters of mentions in context and a hierarchy over them, as shown in Figure 1. The corpus is drawn from computer science papers, and the concept mentions are *methods* and *tasks* from across CS. To build SCICO, we develop a new candidate generation approach built on three resources: a low-coverage KB, a noisy hypernym extractor, and curated candidates. We evaluate strong baseline models, finding that a cross-encoder model addressing coreference and hierarchy jointly outperforms others.[1]

**Our main contributions include:**

- We formulate the novel task of hierarchical cross-document coreference (H-CDCR), and explore it within scientific papers.
- We release SCICO, an expert-annotated dataset 3X larger than the prominent ECB+ CDCR dataset.
- We build a model for H-CDCR that outperforms multiple baselines while leaving much room for future improvement.

## 2. The H-CDCR Task

### 2.1 Problem Formulation

Our goal is to induce clusters of contextualized mentions that corefer to the same concept, and to infer a hierarchy over these concept clusters. Formally, we are given a set of documents $\mathcal{D}$ from a

---

1. SCICO, code and models are available at https://scico.apps.allenai.org/

| | | | |
|---|---|---|---|
| **Diversity** | **Title: Conditional Image Synthesis With Auxiliary Classifier GANs (2017)**
...assessing the discriminability and diversity of *class-conditional image synthesis* ... | $\Longleftrightarrow$ | **Title: Mode Seeking Generative Adversarial Networks for Diverse Image Synthesis**
... for *categorical image generation*, we apply the proposed method on DCGAN using CIFAR-10 dataset... |
| | **Title: Detecting anomalous and unknown intrusions against programs**
... approaches to *detecting computer security intrusions* in real time are misuse detection and... | $\Longrightarrow$ | **Fuzziness based semi-supervised learning approach for intrusion detection system**
*Countering cyber threats*, especially attack detection, is a challenging area of research in information... |
| **Ambiguity** | **Title: On Generalized and Specialized Spatial Information Grids**
...spatial data acquisition, analysis, *information extraction* and knowledge discovery ... | $\Longleftrightarrow$ | **Title: Query selection for automated corpora construction with a use case in food-drug interactions**
... building a high-coverage corpus that can be used for *information extraction* ... |
| | **Title: Semi-Supervised Semantic Role Labeling with Cross-View Training (EMNLP 2019)**
...SRL model can leverage unlabeled data under the *cross-view training* modeling paradigm... | $\Longleftrightarrow$ | **Title: Learning Mid-level Filters for Person Re-identification (CVPR 2014)**
... a *cross-view training* strategy is proposed to learn filters that are view-invariant and discriminative... |

Table 1: **Examples from SCICO**. Scientific concepts exhibit lexical diversity and ambiguity. For example, *information extraction* can refer to the NLP task, or to extracting spatial information from grids; and *cross-view training* can refer to a computer vision technique or a semi-supervised model.

diverse corpus. We assume each $d \in \mathcal{D}$ comes annotated with *mentions* (spans of text, see Table 1) denoting concepts. Denote by $\mathcal{M}_d = \{m_1, m_2, ..., m_n\}$ the set of mentions in document $d$ and by $\mathcal{M}$ the set of mentions across all $d \in \mathcal{D}$.

The first goal, similar to cross-document coreference resolution, is to cluster the mentions in $\mathcal{M}$ into disjoint clusters $\mathcal{C} = \{C_1, ..., C_t\}$, with each cluster consisting of mentions $\{m | m \in C_i\}$ that all refer to the same underlying concept (see Figure 1). To account for the difficulty in precisely delineating the "borders" of extremely fine-grained concepts in scientific literature, clusters are allowed to include subtle variations around a core concept (e.g., *CRF model*, *CRF tagger*).

The second goal is to infer a hierarchy over clusters. Define a graph $G_\mathcal{C} = (\mathcal{C}, \mathcal{E})$, with vertices representing $\mathcal{C}$ (mention clusters), and directed edges $\epsilon_{ij} \in \mathcal{E}$, each edge representing a hierarchical relation between clusters $C_i$ and $C_j$ which reflects *referential hierarchy*. A relation $\mathcal{C}_1 \rightarrow \mathcal{C}_2$ exists when the concept underlying $\mathcal{C}_2$ entails a reference to $\mathcal{C}_1$. For example, a text that mentions the concept "BERT model" also (implicitly) invokes several other concepts ("neural language model", "neural nets", "NLP") but not others ("robotics", "RoBERTa model"). In section 3.3, we ground this entailment definition with a faceted search application to assist the annotation.

Put together, the goal in our **Hierarchical Cross Document Coreference** (H-CDCR) task is: Given documents $\mathcal{D}$ and mentions $\mathcal{M}$, construct clusters $\mathcal{C}$ and a hierarchy graph $G_\mathcal{C} = (\mathcal{C}, \mathcal{E})$ by learning from a set of $N$ examples $\{(\mathcal{D}^k, \mathcal{M}^k, G_\mathcal{C}^k)\}_{k=1}^N$. In our experiments, we focus on mentions referring to *tasks* and *methods* in computer science papers (see examples in Table 1 and §3).

## 2.2 Evaluation Metrics

Comparing an extracted set of mention clusters and hierarchies to a gold standard for evaluation is non-trivial. Evaluation metrics for coreference resolution (e.g., MUC, $B^3$, CEAFe, and LEA) do not involve relations between clusters, and models for inferring hierarchical relations such as hypernymy do so over pairs of *individual* terms. Therefore, in addition to reporting established metrics for coreference resolution, we devise two specific metrics for our novel unified task.

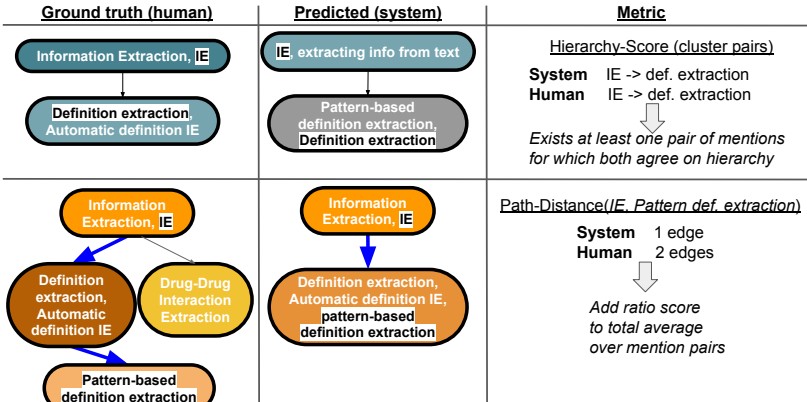

Figure 2: **Evaluation Metrics Examples**. (Top row) Cluster-level hierarchy score. Both agree that *IE* is a parent of *definition extraction*, hence the model is rewarded to avoid doubly penalizing coreference mismatches. (Bottom row) Path-distance score. Graph distances between mention pairs are compared between human and model.

**Cluster-level Hierarchy Score**   In H-CDCR, hierarchical relations are defined over *clusters* of mentions. This complicates the evaluation of the hierarchy, since a system may output a different set of clusters from the gold due to coreference errors. Our cluster-level score is intended to evaluate hierarchical links without penalizing coreference errors a second time. Let $\mathcal{C}_P^S \to \mathcal{C}_C^S$ be a hierarchical link output by the system between a parent cluster $\mathcal{C}_P^S$ and a child cluster $\mathcal{C}_C^S$. We define this link to be a true positive *iff* there exists some pair of mentions in these two clusters that are also in a parent-child relationship in the gold hierarchy. That is, it is positive *iff* there exist mentions $p \in \mathcal{C}_P^S$ and $c \in \mathcal{C}_C^S$ which are also in two gold clusters, i.e. $p \in \mathcal{C}_P^G$ and $c \in \mathcal{C}_C^G$, such that $\mathcal{C}_P^G \to \mathcal{C}_C^G$ in the gold.[2] If not, the output link counts as a precision error. We define recall errors analogously, swapping system and gold in the above formalism.

As an example, see Figure 2, where $p = IE$ and $c = $ *Pattern-based definition extraction*. The metric treats this link as a true positive, since there is at least one pair of mentions in the gold over which the same hierarchy relation holds. Requiring at least one pair, rather than greater overlap, is intended to maximally decouple the coreference and hierarchy penalization. We also report results when requiring a more conservative 50% overlap, to examine robustness to this choice.

**Path-Distance Score**   The cluster-level hierarchy score does not take into account the degree of separation between concepts in the graph. We devise a score that does, following previous work on graph-based semantic distances [Lee et al., 1993].

Here, we consider for each pair of mentions $(m_1, m_2)$ the number of cluster-level directed edges needed to traverse from $m_1$ to $m_2$, plus one (mentions in the same cluster have a distance of 1). For each mention pair, we compute this distance for both the ground-truth and the model (see Figure 2), and the ratio between the smaller of the two distances and the larger. For pairs that are disconnected in the gold or the system but not both, we treat the ratio as zero, and pairs disconnected in both are ignored. The ratios are averaged to obtain the final score.

---

2. We also apply transitive closure, adding edges between pairs of clusters with *indirect* hierarchical relations (e.g., across the first and last nodes in the chain *computer vision* $\to$ *image synthesis* $\to$ *categorial image synthesis*), following common practice in hierarchy prediction [Li et al., 2018].

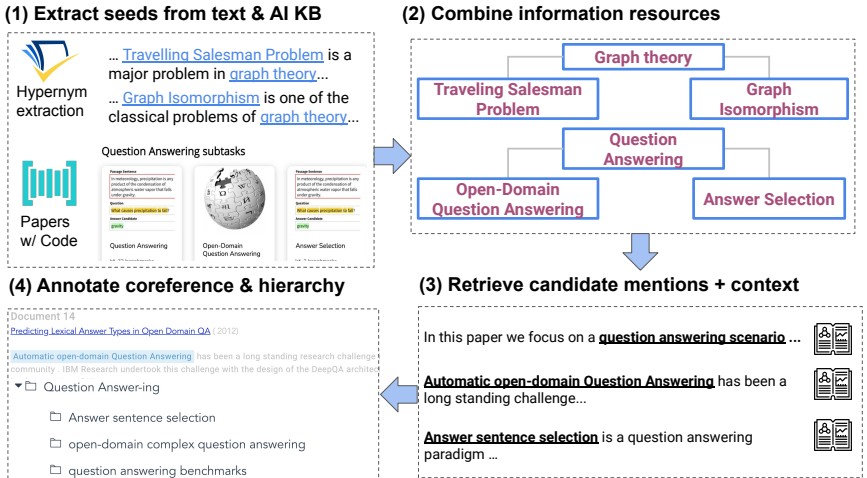

Figure 3: **Overview of the data collection**. (1) We bootstrap data collection by using PwC and a hypernym extractor. (2) The outputs from these resources are combined, forming groups of interrelated concepts used for candidate generation. (3) We retrieve candidates (mentions in context) based on the constructed concept groups. (4) Expert annotators use an interface built for this task.

More formally, let $W$ be the union of mention pairs that have a path in the gold or system cluster-graph. The path-distance score is defined as follows:

$$\frac{1}{|W|} \sum_{i,j \in W} \frac{min(p_{\text{sys}}(i,j), p_{\text{gold}}(i,j)) + 1}{max(p_{\text{sys}}(i,j), p_{\text{gold}}(i,j)) + 1} \tag{1}$$

where $p_{\text{gold}}(i,j)$ and $p_{\text{sys}}(i,j)$ are the distances between the mentions $i$ and $j$ in the gold and the system, respectively.

This metric gives partial credit for similar but not exactly matching graphs, e.g., a predicted parent-child relation is not considered a complete error if the gold specifies a coreference relation (the ratio is 1/2). This can help in ambiguous cases; e.g., whether *CRF model* is a parent of *CRF tagger* or belongs in the same cluster.

## 3. SciCo Data Construction

Obtaining annotations for H-CDCR is non-trivial; showing annotators all possible pairs of documents is not feasible, and presenting randomly drawn mentions is ineffective as they will rarely be related. We thus follow work on data collection for coreference tasks [Cybulska and Vossen, 2014, Jain et al., 2020, Ravenscroft et al., 2021, Eirew et al., 2021] and bootstrap with existing resources.

Our process relies on two primary ingredients: a large corpus of mentions in papers, and a set of seed taxonomies that we leverage to find mentions that are likely to be coreferring or have hierarchical relations. Annotators are then asked to build clusters of mentions and hierarchical relations between them (as in Figure 1). Below, we describe these two ingredients and how we use them for candidate selection. Our overall data construction approach is illustrated in Figure 3.

### 3.1 Documents, Mentions and Seed taxonomies

We populate our dataset of papers and mentions from two data sources: *(i)* a large corpus of 10M CS abstracts from [Lo et al., 2020] and *(ii)* 438 full-text AI papers from SciREX [Jain et al., 2020]. For the 10M abstracts, we extract mentions referring to methods and tasks, using the DyGIE++ IE model [Wadden et al., 2019] trained on SciERC [Luan et al., 2018]. SciREX has the advantage of introducing mentions from full paper texts, vetted by a human annotator for quality.

We select mention sets from our corpus that are suitable for annotators to label. This requires identifying subsets that are likely to contain coreferent or hierarchically-related mentions. We bootstrap such subsets with existing resources described briefly below (see more details in Appendix A).

**High-precision, low-coverage AI KB.** Papers With Code[3] (PwC) is a public resource that maintains a hierarchical KB of ML related tasks, methods, datasets and results. For example, subtasks of Image Classification include *displaced people recognition* and *satellite image classification*.

**Corpus-level hypernym extraction.** To form a higher-coverage resource, we extract a broad (lower-precision) hierarchy of tasks and methods. Specifically, we extract all hypernym relations from the 10M CS abstracts using the DyGIE++ model [Wadden et al., 2019] trained on SciERC. The model extracts hypernym relations between uncanonicalized mentions that appear in the same sentence. We form a hierarchy across the entire corpus by aggregating hyponyms of the same concept string across all papers (see Appendix A.1 for technical details).

**Curated list of lexically diverse candidates.** Automatically collecting lexically diverse mentions is challenging; while our two automated approaches capture such examples, they also capture many lexically *similar* mentions, a common issue in coreference datasets (§5.2). To increase the diversity of SCICO and enable more transparency in evaluation (§5.2), we enrich SCICO with a manually-curated collection of 60 groups of closely related but lexically diverse concepts (400 in total). For example, one group contains {*deep learning*, *neural models*, *DNN algorithms*}, another includes {*class imbalance*, *skewed label distribution*, *imbalanced data problem*}.

### 3.2 Candidate Retrieval

We now turn to how we use the resources described in the previous section in order to select candidate mentions for annotation. Careful candidate generation is often necessary when collecting data for cross-document coreference [Cybulska and Vossen, 2014, Ravenscroft et al., 2021], since the vast majority of random mention pairs are easy negatives. Following standard CDCR terminology, we refer to a batch of candidate mentions to be annotated together as a "topic". Given a topic, annotators are asked to form coreference clusters of mentions and label hierarchical relationships between clusters. Restricting the number of mentions in a topic is necessary in order to make the quadratic complexity of the task tractable.

We form topics of mentions (Figure 3 (3)) as follows. First, we merge the PwC and hypernyms graphs (de-duplicating nodes) and form groups of candidate concepts where each group consists of a single parent concept and its children. We add our manually-curated groups to this set. We then form a topic from each group by matching each of its concepts against our large-scale corpus (§3.1). Specifically, we retrieve the most similar mentions to each concept, in terms of cosine similarity of embeddings output by a neural model (details in Appendix A.2). The union of the retrieved mentions (and their contexts) for a given group form a single topic in our data set (Figure 3 (4)). The mention

---

3. https://paperswithcode.com

retrieval step enriches SCICO with ambiguous cases (e.g., references to *information extraction* with very different meanings), as well as fine-grained variants of concepts.

### 3.3 Data Annotation

Dataset annotation for our task is challenging and requires knowledge in computer science research. We hired 4 PhDs and graduate students in CS through UpWork[4], all authors of at least two scientific publications. Annotators were paid $20-$30 per hour (2-3 topics per hour depending on their size).

In addition to guidelines and tutorials, we also guide annotators to consider the faceted query by example [Mysore et al., 2021] application discussed in the Introduction, to help resolve ambiguity. As a concrete example consider two mentions, $m_1 = ELMo\ model$ and $m_2 = ELMo\ embedding$. Searching for one and retrieving the other should usually be acceptable (indicating coreference). By contrast, *Penn TreeBank POS tagging* should be annotated as a child of *POS Tagging*, since when issuing a faceted query for the PTB variant of POS tagging, in most cases we would not wish to see a list inundated with variants such as POS tagging in tweets or different languages; conversely, searching for *POS Tagging* should show a hierarchy of subsumed concepts, including specific variants. While this framing still leaves some room for subjectivity, we embrace it rather than attempt to formulate many complex rules with inevitably limited coverage. To ensure quality we also provided extensive feedback after the first round of annotation.

We provide annotators with a sample of candidate mentions together with their surrounding context and some metadata of the paper such as title, venue, year and the link to the paper itself, as shown in Figure 3 (4). To annotate SCICO, we extend CoRefi [Bornstein et al., 2020], a recent tool for coreference annotation, by enabling annotation of cluster hierarchies and disaplying metadata (see Appendix A.3 for more details). Annotators are asked to annotate both the clusters and relations between the clusters. We repeat this process for each topic in our pool.

### 3.4 SCICO Properties

Table 2 shows summary statistics of SCICO. Notably, SCICO includes over 26K mentions across about 10K clusters and 6K hierarchical relations. SCICO is 3 times larger than prominent CDCR benchmark ECB+ [Cybulska and Vossen, 2014]. Mentions in SCICO are taken from over 20K scientific documents covering diverse CS concepts, larger than ECB+ by an order of magnitude. The average number of connected components across topics is 6.8, and the average depth of the maximal component (tree) is 3.5.

|             | Train | Validation | Test  | Total |
|-------------|-------|------------|-------|-------|
| # Topics    | 221   | 100        | 200   | 521   |
| # Documents | 9013  | 4120       | 8237  | 20412 |
| # Mentions  | 10925 | 4874       | 10423 | 26222 |
| # Clusters  | 4080  | 1867       | 3711  | 9538  |
| # Relations | 2514  | 1747       | 2379  | 5981  |

Table 2: SCICO statistics.

To measure agreement, all annotators labeled the same 40 randomly chosen topics (groups of candidates, totalling about 2200 mentions). Following common evaluation practices in coreference resolution, we measure micro-averaged pairwise agreement (denoted AVG), considering one annotation as gold, and the other as predicted (metrics are symmetric). We also measure the average maximal pairwise IAA, as an "upper bound" measure of human performance (MAX-micro). Finally we compute the maximal IAA scores for each topic, and average those 40 scores (MAX-macro).

---

4. http://upwork.com/

Models for coreference resolution are traditionally evaluated using different metrics, MUC [Vilain et al., 1995], B$^3$ [Bagga and Baldwin, 1998], CEAFe [Luo, 2005] and LEA [Moosavi and Strube, 2016], while the main evaluation is CoNLL F1, which is the average F1 of MUC, B$^3$ and CEAFe. For CoNLL F1, AVG is $82.7(\pm2.5)$, MAX-micro is $85.8$ and MAX-macro $90.2$.[5] See more coreference metrics in Table 7 in Appendix C. For our cluster-hierarchy F1 metric we get AVG of $68.9(\pm2.2)$, MAX-micro of $72.1$ and MAX-macro of $82.3$. The path-distance agreement scores are, respectively, $64.5(\pm3.5)$, $70.0$, and $78.4$. Importantly, as we will see in §5 these IAA rates are substantially higher than our best model's performance, leaving much room for future modelling.

## 4. Models

We now present our models for H-CDCR. We start by presenting baselines that separately predict coreference clusters and the hierarchy between them (§4.1). Then, we describe a model that optimizes both subtasks simultaneously (§4.2).

### 4.1 Baseline Models

**Coreference Model**   We use a recent state-of-the-art cross-document coreference model [Cattan et al., 2021a] for predicting clusters of mentions (denoted by **CA**; see details in Appendix B.1).

We experiment with several variants of this model. First, to evaluate how SCICO differs from existing CDCR datasets, we explore two versions trained only on external CDCR resources: **CA$_{\text{News}}$** trained on ECB+ [Cybulska and Vossen, 2014] and **CA$_{\text{Sci-News}}$**, trained on CD$^2$CR [Ravenscroft et al., 2021] that includes coreference annotation between a single news article and linked scientific paper with mentions extracted using NER. Next, we train the model on SCICO. In addition to the RoBERTa (**CA$_{\text{RoBERTa}}$**) pretrained language model used in [Cattan et al., 2021a], we explore models specialized to our scientific domain: CS-RoBERTa (**CA$_{\text{CS-RoBERTa}}$**) [Gururangan et al., 2020] and SciBERT (**CA$_{\text{SciBERT}}$**) [Beltagy et al., 2019].

**Hierarchy Model**   We consider a referential hierarchical relation between concepts $x$ and $y$ as an *entailment* relation between $y$ and $x$ [Glockner et al., 2018].[6] We use a model for textual entailment (RoBERTa-large-MNLI) [Liu et al., 2019], representing each cluster $\mathcal{C}$ by the concatenation of its mentions (without context) `[CLS]` $\mathcal{C}_k$ `` $\mathcal{C}_j$ ``, and running RoBERTa-large-MNLI on all $n(n-1)$ cluster pairs $(\mathcal{C}_k, \mathcal{C}_j)$. The `[CLS]` embedding is fed into an output layer for entailment classification. To avoid creation of cyclical graphs, we adopt a greedy approach adding relations iteratively, starting from the highest entailment scores, discarding relations creating cycles. We apply this model to clusters obtained by each baseline.

### 4.2 Unified Model for H-CDCR

Deciding whether two mentions refer to the same concept or have a hierarchical relation is sometimes non-trivial (e.g *POS Tagging* $\rightarrow$ *PTB POS Tagging*, but *artificial neural networks* and *neural networks* refer to the same concept). Therefore, we develop a unified model by formulating our task as multiclass classification, where each mention pair $(m_1, m_2)$ can be assigned into four classes (0)

---

5. For comparison with OntoNotes [Pradhan et al., 2012], we also measure the average MUC F1 and report a score of 89.6, 2.2 F1 points higher than Ontonotes.

6. Indeed, "John ate an apple" *entails* "John ate a fruit" because referring to "apple" entails reference to "fruit".

$m_1, m_2$ corefer (1) $m_1 \rightarrow m_2$ (2) $m_2 \rightarrow m_1$ or (3) not related. Consider a topic (pool of candidates, §3.2), with a set of mentions $\mathcal{M}$. During training, we learn a pairwise scorer $f(\cdot, \cdot)$ by minimizing:

$$L = -\frac{1}{N} \sum_{\substack{m_1, m_2 \in \mathcal{M} \\ m_1 \neq m_2}} y \cdot log(f(m_1, m_2)) \qquad (2)$$

where 1-hot $y$ is one of the four classes and $N$ is the number of training pairs. For $f(\cdot, \cdot)$ we use Longformer [Beltagy et al., 2020], a transformer-based language model for processing long sequences so that we can encode pairs of full paragraphs. We also use CDLM, a recent variant of Longformer pre-trained for cross-document tasks [Caciularu et al., 2021].

During pretraining, Longformer and CDLM apply *local attention* — attention only to tokens in a fixed-sized window around each token. When fine-tuning on a specific task, *global attention* — attention to all tokens in the sequence — can be assigned to a few target tokens to encode global information. Following Caciularu et al. [2021], we take each mention and its corresponding paragraph, inserting mention markers <m> and </m> surrounding the mention to obtain a mention representation. For CDLM, we add the document markers <doc-s> and </doc-s> surrounding each document. Then, we concatenate the representations of $m_1$ and $m_2$ separated by a separator token , and add a [CLS] token at the beginning. We assign *global attention* to the [CLS] and the mention markers of the two mentions. The [CLS] vector is finally fed into a linear layer $W^{d \times 4}$, for fine-tuning the model.

At inference time, we build clusters using agglomerative clustering over predicted coreference scores, in the same way as in the baseline described above (§4.1). Then, for each pair of clusters $(\mathcal{C}_1, \mathcal{C}_2)$, we aggregate cross-cluster mention-pair predictions for hierarchical relations as follows. Given a pair $(m_1, m_2)$ where $m_1 \in \mathcal{C}_1$ and $m_2 \in \mathcal{C}_2$, we compute the probability score for $m_1$ being a child of $m_2$, and define the score of $C_1$ being the child of $C_2$ as the average of all pairwise scores for all $\{(m_i, m_j) | m_i \in \mathcal{C}_1, m_j \in \mathcal{C}_2\}$:
$s(\mathcal{C}_1, \mathcal{C}_2) = \frac{1}{|\mathcal{C}_1| \cdot |\mathcal{C}_2|} \sum_{m_1 \in \mathcal{C}_1} \sum_{m_2 \in \mathcal{C}_2} f_{\text{is-child}}(m_1, m_2)$.

|  | Coreference | Hierarchy | | Path |
|---|---|---|---|---|
|  | CoNLL F1 | F1 | F1-50% | Ratio |
| IAA (AVG) | 82.7 | 68.9 | 62.8 | 64.5 |
| IAA (MAX-Macro) | 90.2 | 82.3 | 77.7 | 78.4 |
| CA$_{\text{News}}$ | 52.4 | 37.1 | 13.0 | 24.1 |
| CA$_{\text{Sci-News}}$ | 43.5 | 29.2 | 12.3 | 21.6 |
| CA$_{\text{SciCo}}$ | 55.2 | 23.7 | 15.8 | 21.2 |
| CA$_{\text{SciCo}}$ + CS-RoBERTa | 57.4 | 23.5 | 16.1 | 23.6 |
| CA$_{\text{SciCo}}$ + SciBERT | 66.8 | 23.8 | 17.8 | 28.4 |
| Unified$_{\text{Longformer}}$ | **77.2** | 44.5 | **36.1** | **47.2** |
| Unified$_{\text{CDLM}}$ | 77.0 | **44.8** | 35.5 | 45.9 |

Table 3: **Model results**. We evaluate strong CDCR baselines, and a unified multiclass model for H-CDCR that outperforms the baselines.

To avoid cycles we apply the same greedy approach as in the baseline models (§4.1) and stop when the hierarchy score is below a tuned threshold. Full implementation details and hyperparameters are described in Appendix B.2.

## 5. Experimental Results

Table 3 presents the results of the baseline models as well as our unified solution. Results of coreference are reported using the CoNLL F1 metric. We also report performance with our two metrics

introduced earlier (§2.2). See Appendix C for all coreference metrics, as well as recall and precision for the hierarchy score.

The results show that training a model to predict both coreference and hierarchy in a multiclass setup outperforms baselines by a large margin across all metrics. In terms of coreference, using the **CA** baselines (§4.1) trained on external datasests leads to the lowest results. Training the same model on SciCo boosts results, with a significant boost from SciBERT [Beltagy et al., 2019]. Focusing on hierarchy, we examine results in terms of our cluster-level hierarchy metric (§2.2) which is designed to not doubly-penalize coreference errors, and the path-based metric. For all baselines, we use a state-of-art pre-trained entailment model [Liu et al., 2019] (see §4) for predicting relations between clusters, which leads to poor results in comparison to the unified model.

## 5.1 Ablation Study

We conduct an ablation study examining our unified formulation and utility of wider contexts. To ablate the unified model (multiclass), we train two models using the same architecture (§4.2): one for coreference only, the other for hierarchy only. In the former, we consider hierarchical relations as negative pairs, and train a binary classification model with classes (0) unrelated and (1) coreference. The hierarchy-only

|  | Coreference | Hierarchy |  | Path |
|---|---|---|---|---|
|  | CoNLL F1 | F1 | F1-50% | Ratio |
| − unified | 77.1 (−0.1) | 41.6 (−2.9) | 32.3 (−3.8) | 44.2 (−3.0) |
| − paragraph | 77.0 (−0.2) | 43.0 (−1.5) | 33.9 (−2.2) | 45.7 (−1.5) |

Table 4: **Ablation results**. Parentheses show the relative drop in performance. Both large context and the unified approach contribute to the scores.

model has classes (0) unrelated, (1) parent-child and (2) child-parent. During inference, we apply the same approach as in §4.2 for creating clusters and inferring cluster hierarchy.

As shown in Table 4, the unified model outperforms the disjoint approach in the cluster-hierarchy and path-based scores, indicating the utility of optimizing for both tasks simultaneously. Learning a single model also results in half the number of parameters than in the disjoint approach, while achieving substantially better results. Finally, using only a mention's sentence as context (instead of full paragraph) also leads to a performance drop.

## 5.2 Lexical Diversity Impact on Coreference

We explore how coreference performance correlates with lexical diversity (and ambiguity). We examine a simple baseline that uses the Levenshtein edit distance in agglomerative clustering. Surface-form matching baselines are known to have comparatively fair performance in CDCR datasets [Barhom et al., 2019, Eirew et al., 2021]. We take the bottom 10% and bottom 20% of topics ranked by the baseline's CoNLL F1 (20 and 40 topics, respectively). We also examine the set of manually curated lexically diverse topics (§3.1).

| Test subset | Unified | Edit dist. |
|---|---|---|
| Full | 77.2 | 53.3 |
| Lowest 10% (edit dist.) | 64.4 | 27.2 |
| Lowest 20% (edit dist.) | 69.4 | 34.6 |
| Curated (lexically diverse) | 67.1 | 35.0 |

Table 5: **CoNLL F1 results by lexical diversity**. We take the bottom 10% (20%) topics ranked an edit-distance baseline's CoNLL F1, and also examine manually curated lexically diverse topics (§3.1). Results indicate SciCo contains subsets with varying levels of lexical diversity that correlate with coreference difficulty.

In Table 5, we report micro-averaged results for each subset. Our unified model does substantially worse than its overall performance (e.g., the model does not identify coreference between *scientific paper analysis* and *scholarly document analysis*, *manual annotation* and *human labeling*). We also check that the edit-distance baseline does not correlate with inter-annotator agreement, to ensure IAA rates reported earlier can serve as a measure of human performance across levels of diversity. Using Pearson and Spearman correlation tests, we confirm this hypothesis (p-values of 0.10, 0.21). These results indicate SCICO contains subsets with varying levels of lexical diversity that correlate with coreference difficulty, requiring richer information and models to resolve.

## 6. Related Work

**Entity coreference resolution**    Entity coreference work focuses on mentions within a single document (WD), while cross-document (CD) work focuses on coreference between *events* in the news [Cybulska and Vossen, 2014, Vossen et al., 2018]. In the science domain, some work has been done on entity coreference in the WD setting [Luan et al., 2018, Jain et al., 2020]. Unlike previous work, in this paper we considered CDCR jointly with inferring hierarchy, for a domain with abstract technical concepts that are nested in many levels of granularity that can be hard to tell apart.

**Entity linking**    Entity linking (EL) involves linking mentions of entities to knowledge base (KB) entries. In science, KBs are often scarce and highly incomplete [Hope et al., 2021]. In our work, we used a low-coverage KB to bootstrap data collection and did not assume to have a KB during training or inference. Recent work [Angell et al., 2021] has shown that clustering WD mentions can boost EL in biomedical papers; our work on CD mentions could help by pooling information from diverse contexts, and potentially detect entities missing from KBs [Lin et al., 2012, Wu et al., 2016].

**Taxonomy construction**    Most work in this area generates a graph where nodes are *single* terms representing a concept (e.g., hypernym-hyponym pairs extracted from single sentences) [Hearst, 1992, Roller et al., 2018]. This approach does not resolve ambiguity of identical surface forms (e.g., *information extraction*) and lexical diversity of concepts. Recent work in the data mining community [Shang et al., 2020] focused on unsupervised construction of taxonomies, with clusters of uncontextualized terms given as input. Our work can potentially help these and related applications [Poon and Domingos, 2010] by introducing context and supervision.

**Word sense induction**    A related line of work focuses on learning to induce multiple senses of words from text, to capture polysemy and resolve ambiguity. Such work typically employs uncontextualized word embeddings [Athiwaratkun and Wilson, 2017, Arora et al., 2018] or phrase-level embeddings [Chang et al., 2021]. Relatedly, Shwartz and Dagan [2016] label pairs of *words* for equivalent senses depending on context, in general language rather than abstract technical concepts.

## 7. Conclusion

We present SCICO, a dataset for a novel task of hierarchical cross-document coreference resolution (H-CDCR) in the challenging setting of scientific texts. SCICO is annotated by domain experts and is three times larger than comparable datasets from the news domain. We evaluate strong baseline models on our data. A joint approach that infers both coreference and hierarchical relationships in the same model outperforms multiple baselines, but leaves substantial room for improvement.

**Acknowledgments:**    Many thanks to anonymous reviewers. This project is supported in part by NSF Grant OIA-2033558, NSF RAPID grant 2040196, and ONR grant N00014-18-1-2193.

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

## Appendix A. Data Collection

### A.1 Hypernym extraction details

As mentioned in the paper (§3.1), we build a large hypernym graph based on the extracted hypernym relations at the sentence level. For example, given the sentences (from two different papers): *"The Travelling Salesman Problem (TSP) is one of the major problems in graph theory."* and *"Graph isomorphism is a major problem in graph theory."*, the Dygie++ model identified the relations graph theory → Travelling Salesman Problem (TSP) and graph theory → Graph isomorphism. In our corpus-level hypernym extraction procedure, we use standard surface-level string normalization to unify mentions across the corpus (removing punctuation, lemmatizing, lowercasing and the Levenshtein edit-distance with threshold 0.8), resulting in distinct 250K tasks and 1.2M methods with 340K and 1.4M hierarchical edges between them respectively. We then sample parent-child mentions and siblings to form our candidates (see Figure 3 (1-2)).

In addition to surfacing candidates for the referential hierarchy, another significant advantage of this resource is indeed one of its main weaknesses (discussed in §6): it produces coreferring mentions of the same concept in the form of hyponyms of a shared parent, which we use to enrich SciCo for the CDCR task (e.g., two different papers mentioning *image synthesis* or *image generation* as children of *computer vision tasks*).

While this automatically extracted hierarchy has broad coverage, it suffers from noise.[7] To reduce noise, an annotator who did not take part in the final data annotation (§3.3) was given a set of generated candidates and post-processed to filter overly generic or noisy spans.

### A.2 Mention Retrieval Details

Formally, denote by $\mathcal{S} = (V, E)$ the union of the PwC and hypernym graphs. For a parent concept $p \in V$, let $\{c\}_p$ be its set of direct descendants (children). We take the parent vertex $p$ and its child vertices $\{c\}_p$, and add the 60 curated groups to this set, to form a complete set of candidates we denote by $C_p$.

For the sources making up $\mathcal{S}$, the concept names in PwC and the curated lists are detached from any specific paper context, which is required for building SciCo. While the extracted hypernym relations do come from specific sentences, using only those contexts as candidates would bias models to focus only on within-document relations. In addition, we wish to diversify our the surface forms of mentions by retrieving subtle variants (e.g., retrieving for *BERT* mentions such as *BERT model*, *BERT architecture*, *BERT-based representation*, etc.).

Thus, we augment each group by retrieving similar mentions from our corpus. Specifically, for each $C_p$ we find similar mentions for each $c \in C_p$. We use an encoding function $f : c \mapsto \mathbb{R}^d$ that maps the surface form of each selected $c$ to $f(c)$, a $d$ dimensional vector.[8] Following the approach of [Reimers and Gurevych, 2019], the encoding function is obtained by fine-tuning a language model pre-trained on computer science papers [Gururangan et al., 2020] on two semantic similarity tasks, STS [Cer et al., 2017] and SNLI [Bowman et al., 2015]. We then apply $f(\cdot)$ to all distinct mentions in our large-scale corpus with over 30M mentions. Finally, we augment each group with the top $K$ highest-scoring mentions by cosine similarity to each $c \in C_p$ in the initial group and take the union

---

7. For example, from the text ... *image information for analysis purposes, such as segmentation, identification ...*, we obtain the hypernym relation (analysis purposes, identification).

8. We also experimented with using the context of the mentions as well, but found this to result in more easy near-exact matches along with more highly noisy ones.

of retrieved results. To manage the scale, we employ a system designed for fast similarity-based search [Johnson et al., 2017].[9] To make sure we sample enough mentions from SciREX despite its comparatively small size, we sample from it with the same proportion as from the 10M abstracts.

### A.3 Annotation Interface

To annotate SciCo, we extend CoRefi [Bornstein et al., 2020] with the ability to annotate hierarchy of clusters.[10] The hierarchy is kept in sync with the clusters – any modification of the clusters (e.g., merging two clusters) affects automatically the hierarchy (e.g., unifying their children). Therefore, annotators can annotate both the clusters and the hierarchy at the same time or alternate between them. In addition, annotators can (and are encouraged to) add some notes in the hierarchy to help them distinguish between ambiguous concepts. As shown in Figure 3, we extend CoRefi by displaying metadata for each paper, including the link to the Semantic Scholar URL of the paper. This additional context is often helpful in order to annotate complex cases. For example, as shown in Table 1, for resolving the ambiguity between two mentions of *cross-view training*, it may be useful to see that one paper was published a few years before the other one coined its own *cross-view training* as a name for their semi-supervised model.

## Appendix B. Models

### B.1 Cattan et al. [2021a]'s CDCR model

Here, we describe the cross-document resolver of [Cattan et al., 2021a] that we use as baseline for SciCo (§4.1). This model is based on contextualized token and span representations. For a document $d$, we are given a sequence of tokens $\mathbf{x} = \{x_1, x_2, \ldots, x_T\}$ where $T$ is the length of the document. We first obtain a contextualized embedding of each $x_t \in \mathbf{x}$ using the RoBERTa-large pre-trained language model [Liu et al., 2019]. Each mention span $i$ is a contiguous subsequence of $\mathbf{x}$, denoted $\mathbf{x}_i = \{x_1^{(i)}, x_2^{(i)}, \ldots, x_S^{(i)}\}$. Let $\hat{\mathbf{x}}_i = \{\hat{x}_1^{(i)}, \hat{x}_2^{(i)}, \ldots, \hat{x}_S^{(i)}\}$ be the sequence of embedded tokens in span $i$, where $\hat{x}_t^{(i)}$ is the embedding of token $t$ in span $i$. Then, similarly to Lee et al. [2017], each span $i$ is represented (3) by the concatenation of embeddings of the boundary tokens in the span ($\hat{x}_1^{(i)}$ and $\hat{x}_S^{(i)}$), an attention-weighted sum of the token embeddings in span $i$ (Attn($\hat{\mathbf{x}}_i$)), and a feature vector denoting the span width (number of tokens).

$$g_i = [\hat{x}_1^{(i)}, \hat{x}_S^{(i)}, \text{Attn}(\hat{\mathbf{x}}_i), \phi(i)] \tag{3}$$

Given a pair of contextualized span embeddings $g_i$ and $g_j$ from two different documents, we feed them to a pairwise scorer (4) in the form of a simple feed-forward network that receives as input the concatenation of two span representations and their element-wise product, and outputs the likelihood that the two mentions corefer.

$$s(i, j) = \text{FFNN}([g_i, g_j, g_i \circ g_j]) \tag{4}$$

At inference time, the coreference clusters are formed by applying agglomerative clustering [Ward Jr, 1963] over a pairwise distance matrix populated with scores $s(i, j)$, using the average-linkage cluster merging criterion. Following [Cattan et al., 2021a], this model does not involve

---

9. We filter for mentions with similarity greater than .8, empirically observing sufficient diversity and precision.
10. https://github.com/ariecattan/CoRefi

| Model | Hierarchy | | | Hierarchy 50% | | |
|---|---|---|---|---|---|---|
| | Recall | Precision | F1 | Recall | Precision | F1 |
| CA$_{\text{News}}$ | 43.3 | 32.4 | 37.1 | 21.8 | 9.3 | 13.0 |
| CA$_{\text{Sci-News}}$ | 37.8 | 23.7 | 29.2 | 12.2 | 12.5 | 12.3 |
| CA$_{\text{SciCo}}$ | **45.5** | 16.0 | 23.7 | 17.7 | 14.3 | 15.8 |
| CA$_{\text{SciCo}}$ + CS-RoBERTa | 43.6 | 13.1 | 23.5 | 19.5 | 13.8 | 16.1 |
| CA$_{\text{SciCo}}$ + SciBERT | 41.4 | 16.7 | 23.8 | 27.2 | 13.3 | 17.8 |
| Unified$_{\text{Longformer}}$ | 36.6 | 56.7 | 44.5 | **29.1** | 48.2 | **36.3** |
| Unified$_{\text{CDLM}}$ | 36.4 | **58.2** | **44.8** | 27.8 | **49.1** | 35.5 |

Table 6: Recall, Precision and F1 according to the Cluster-level Hierarchy Score (§2.2) for all models.

fine-tuning of the underlying language model due to prohibitive memory costs — it begins with encoding all documents using an LM, then computes the pairwise scores over all pairs of mention representations. We note that our unified approach is the only model we evaluate that fine-tunes the Longformer on our data.

### B.2 Hyper parameters for the Unified Model

We develop our model in Pytorch [Paszke et al., 2019] and PytorchLightning [Falcon et al., 2019] using the Transformers library [Wolf et al., 2020] and the AdamW optimizer. We train our model on 1 epoch using a batch size of 4 and gradient accumulation of 4 and a learning rate of 1E-5. We conduct our experiments on 8 Tesla V100 32GB GPUS using distributed data parallelism. The training time takes about 2.5 hour for the unified model and inference 25 minutes. Our model includes 148M parameters. We fine-tune the threshold for the agglomerative clustering and the stopping criterion for the hierarchy on the validation set, in a range of {0.4, 0.6} for both, set to 0.6 and 0.4 respectively. We take the values that achieve the best path ratio metric on the validation set.

## Appendix C. Coreference and Hierarchy Results

Table 6 presents the results of all models according to the cluster-level hierarchy score (§2.2) in terms of recall, precision and F1.

Table 7 presents the results of the inter-annotator agreement (IAA), all baseline models as well as the pipeline and unified, according to all common coreference metrics (MUC, B$^3$, CEAFe, LEA and CoNLL F1). We obtain coreference metrics using the python implementation of the standard coreference metrics [Moosavi and Strube, 2016].[11] Following Cattan et al. [2021b], we apply coreference metrics only on non-singleton (gold and predicted) clusters in order to avoid inflated results.

---

11. https://github.com/ns-moosavi/coval/

| | MUC | | | $B^3$ | | | $CEAFe$ | | | LEA | | | CoNLL |
|---|---|---|---|---|---|---|---|---|---|---|---|---|---|
| | R | P | $F_1$ | R | P | $F_1$ | R | P | $F_1$ | R | P | $F_1$ | $F_1$ |
| IAA | - | - | 89.6 | - | - | 81.4 | - | - | 77.1 | - | - | 79.7 | 82.7 |
| CA$_{News}$ | 83.8 | 64.0 | 72.5 | 69.9 | 35.5 | 47.1 | 32.6 | 44.2 | 37.5 | 65.4 | 31.5 | 42.5 | 52.4 |
| CA$_{Sci-News}$ | 75.4 | 65.5 | 70.1 | 66.5 | 24.8 | 36.1 | 17.4 | 41.4 | 24.5 | 63.6 | 22.0 | 32.7 | 43.5 |
| CA$_{SciCo}$ | 54.9 | 81.0 | 65.4 | 40.3 | 73.1 | 52.0 | 48.1 | 48.2 | 48.1 | 38.5 | 69.2 | 47.8 | 55.2 |
| CA$_{SciCo}$ + CS-RoBERTa | 58.4 | 79.7 | 67.4 | 44.8 | 69.9 | 54.6 | 48.9 | 51.3 | 50.1 | 41.1 | 65.9 | 50.7 | 57.4 |
| CA$_{SciCo}$ + SciBERT | 78.0 | 78.8 | 78.4 | 65.6 | 64.6 | 65.1 | 56.4 | 57.2 | 56.8 | 62.4 | 61.3 | 61.9 | 66.8 |
| Unified$_{Longformer}$ | **88.5** | 84.9 | **86.7** | **79.4** | 72.8 | **75.9** | 67.8 | **70.4** | **69.0** | **77.4** | 70.4 | **73.7** | **77.2** |
| Unified$_{CDLM}$ | 87.4 | **85.3** | 86.3 | 77.8 | **73.8** | 75.8 | **68.0** | 70.0 | **69.0** | 75.6 | **71.3** | 73.4 | 77.0 |

Table 7: Coreference results of the IAA, baseline models, pipeline and unified on the SciCo test set according to all coreference metrics: MUC, B$^3$, CEAFe, LEA and CoNLL F1. Since IAA is computed by considering one annotator as gold and the other as system, the IAA is reported only according to the symetric F1.

