# OpenReview forum: "SciCo: Hierarchical Cross-Document Coreference for Scientific Concepts"
_AKBC.ws/2021/Conference — AKBC 2021_

### Official Review · Reviewer_Y8ng · 2021-07-20
**An interesting expansion on cross-document coreference, rigorously annotated dataset and good baselines**

**Rating:** 8
**Confidence:** 4

**Review:**

# General comments
This paper presents a new dataset and baselines for the task of cross-document coreference resolution for concepts in CS papers. In addition to the regular coreference clusters, the authors introduce a hierarchical element, adding parent-child relations between the concepts. Overall, this is a rigorous annotation workflow and the baseline experiments are well designed show the challenging nature of the dataset. I only have a few issues.


# Specific comments/questions
While it is certainly novel in its current formulation (and in the context of scientific concepts), I would argue that the task of hierarchical cross-document coreference has been studied in the past, e.g. the work of Signh et al. (2011) "Large-Scale Cross-Document Coreference Using Distributed Inference and Hierarchical Models".

In Section 3.2 there is a mention of a "parent concept". How is that determined for the automatically generated groups and where is it specified for the curated list presented in Section 3.1?

The example (and justification) given for the referential hierarchy (Section 3.3 and Appendix A.3) isn't entirely convincing. From an information retrieval point of view, I wouldn't mind if a query for Penn TreeBank POS tagging returned documents related to POS tagging in general (in fact it might be more useful to place a specific tagging scheme in the context of its research area).

Why is the Longformer needed in Section 4.2 if the pairwise scorer f(.,.) accepts only the mentions as input? Are the contexts of each also used in the scorer?

# Minor comments
- Flip the order of the presentation of the metrics in Figure 2 to match the order of their introduction in the text.
- I assume the annotators hired (Section 3.3) were from the CS field?
- The CoNLL F1 metric is used in Section 3.4 before its definition in Section 5.
- Are "deep neural networks" and "neural networks" equivalent (as stated in Section 4.2)? I would argue the two terms have exactly the same relationship as "PTB POS tagging" and "POS tagging" (one being a subset of the other).
- For clarity, add the human IAA scores in Table 3.
- The names of the Ablation experiments (Table 4) don't match the names/descriptions used in the text.

---

> ### Author Response · Authors · 2021-07-28
> **Thanks for your review and feedback! We have uploaded a revised version of the paper with your suggestions.**
>
> Thank you for your time reviewing our paper and helpful feedback!
>
> We are encouraged that the reviewer finds interesting our expansion of cross-document coreference resolution with hierarchical relations, along with rigorous annotation workflow and well-designed experiments. See below the answers to your specific questions:
>
>
>
>
> *RE comparison to Singh et al. (2011): “Large-Scale Cross-Document Coreference Using Distributed Inference and Hierarchical Models”.
>
> Singh et al. (2011) address the problem of cross-document coreference resolution (CDCR) on large-scale corpora. To overcome the computation complexity, they propose a hierarchical _model_ where mentions are distributed across several machines (super-entities) and initial coreference clusters are computed in parallel in each machine (sub-entity). These sub-entities are finally merged to form the entity clusters.
> In contrast, we introduce a novel _task_ of hierarchical cross-document coreference resolution, where final coreference clusters have hierarchical relations between them, explicitly annotated in our dataset (e.g., sentiment analysis → aspect-based sentiment analysis).
>
>
>
> *RE “parent concept” for the automatically generated groups and curated list
>
> The “parents" referred to in Section 3.2 are either taken directly from the PwC graph (which has explicit parent links) or from the hypernym links extracted by Dygie++.  For example, given the sentences (from two different papers): “The Travelling Salesman Problem (TSP) is one of the major problems in graph theory” and “Graph isomorphism is a major problem in graph theory”, the Dygie++ model identified the relations graph theory  → Travelling Salesman Problem (TSP) and graph theory → Graph isomorphism.  The curated list does not have explicit parent relationships, and is instead treated as a flat list of concepts.
>
>
>
>
> *RE Penn TreeBank POS tagging vs. POS tagging.
>
> In our referential hierarchy, a concept A is an ancestor of another concept B in the hierarchy if and only if a mention of B is also implicitly a reference to A.  For example, a text that mentions the concept “BERT model” also (implicitly) invokes several other concepts (“neural language model”, “neural nets”, “NLP”, etc.) but not others (“robotics,” “RoBERTa model”). The information retrieval setting In section 3.3 and Appendix A.3 is used for grounding this definition with a faceted search application, used to help annotators reach substantial agreement. We certainly agree that when searching for a specific variant of POS tagging it could also be useful -- in certain use cases -- to view more general POS tagging results as well. In hierarchical faceted search, this information would be presented in the form of concept hierarchies that can be browsed (Marti Hearst, CACM 2006), such that POS tagging would appear as a parent concept for specific variants.
>
> *RE Why is the Longformer needed in Section 4.2 if the pairwise scorer f(.,.) accepts only the mentions as input? Are the contexts of each also used in the scorer?
>
> Yes, as mentioned in 4.2, we use Longformer to encode the mentions with their corresponding paragraphs as context.
>
> *RE: minor comments
>
> We addressed your comments and integrated your suggestions in an updated version of the paper. The hired annotators were indeed from the CS field and we added this detail in the paper.

---

> > ### Comment · Reviewer_Y8ng · 2021-07-30
> > **Thanks for the clarifications**
> >
> > I would like to thank the authors for their clarifications. The distinction regarding Singh's model and your task definition makes sense. I would appreciate if your detailed explanations about my other comments made it into the final version of the paper.

---

### Official Review · Reviewer_EBkC · 2021-07-22
**An interesting reformulation of CDCR accompanied by a robust dataset and models**

**Rating:** 8
**Confidence:** 4

**Review:**

This paper proposes a reformulation of CDCR to include an hierarchical component (H-CDCR), whereby an hierarchy of co-reference clusters is jointly
inferred with the clusters.

Along with the new task, two evaluation metrics are proposed, the cluster-level hierarchy score that evaluates hierarchical links while not penalizing co-reference errors repeatedly (TP when at least one Parent -> Child relation between mentions in common between the system output and the gold cluster hierarchy), the path-based score average path-distance ratios over all pairs of mentions.

The authors propose SciCo, a resource and benchmark for (H-)CDCR thrice the size of the previous reference dataset (ECB+).
The dataset is constructed following a robust protocol that follows established conventions for the annotation of coreference data and that is based on two main components: a large corpus of scientific papers annotated with mentions and a set of seed taxonomies to select mentions that are likely to have hierarchical relations.

The corpus of mentions is build on a corpus of 10M computer science abstracts  + SciREX using a state-of-the-art information extraction model DyGIE++. The model is trained on SciERC, a reference corpus of manually verified mention annotations.

The selection of likely cluster candidates is performed using the papers with code taxonomy, hypernym relations extracted by DyGIE++ (aggregated by concept across all papers). The candidates are augmented with a manually curated list to add diversity in the dataset.
Candidates are grouped into one-parent-multiple-children hierarchies and candidate mentions are retrieved for each group to form the CDCR "topics" using a vector-based retrieval of mentions from the large corpus annotated with mentions.

4 annotators (PhD or graduate) were hired (upwork) to perform the annotations. The corpus shows good agreement (well above system performance, leaving room for improvement).

The authors then perform an evaluation of several baseline systems adapted for the new formulation of the task (SOTA Coreference model and a textual entailment model (for the hierarchy)) and implement their own joint model that significantly outperforms the baseline models.
The joint model casts the association of a pair of mentions as a text classification problem, where two mentions can belong to the following classes: coreference, m1 > m2, m2 > m1, unrelated.

The evaluation is complemented with an ablation study that examines performance with coreference and hierarchy only variants. The impact of lexical diversity is also studied.


Strengths:
 - Well written paper
 - Very strong contributions, both data-wise and methodology-wise

Shortcomings:
 - The paper feels somewhat compressed to fit into the format, some of the elements in the appendices should really be in the paper as the sole descriptions in the paper wouldn't allow to fully reproduce the work. I really needed to read the appendices to fully understand some aspects.




Other comments and questions:

"SciREX has the advantage of introducing mentions from full paper texts, vetted by a human annotator for quality" -> do you mean SciERC?

Formatting issue with the reference for "Hierarchical grouping to optimize an objective function."

About the annotators, it's common to report whether the annotators were compensated fairly for the work and the associated workload.

The textual description for the path-based measure is somewhat difficult to follow, a formula or small algorithm that summarizes the computation would be a welcome addition.

---

> ### Author Response · Authors · 2021-07-28
> **Thanks for your review and feedback! We have uploaded a revised version of the paper and moved important details from the appendices to the main paper.**
>
> Thank you for your time reviewing our paper and helpful feedback!
>
> We are encouraged that the reviewer finds strong contributions in our paper, both data-wise and methodology-wise. See below the answers to your questions:
>
> *RE "SciREX has the advantage of introducing mentions from full paper texts, vetted by a human annotator for quality" -> do you mean SciERC?
>
> We actually meant SciREX. This sentence explains why we chose SciREX as an additional data source for SciCo. In the large corpus of 10M abstracts (Lo et al., 2020), mentions appear only in the abstract and they are automatically extracted using the Dygie++ model. Using SciREX as a data source enriches the diversity and quality of SciCo because (1) mentions also appear in the rest of the paper and (2) mention spans were annotated by human annotators.
>
> *RE compensation of the annotators
>
> As mentioned in Appendix A.3, annotators were paid with an hourly reward between 20 and 30 dollars, where each topic took an average of 30 minutes. We have moved this information in the main paper.
>
> Finally, we added a formula for the path-distance metric. Also, given the additional page, we moved some important details from the appendices to the main paper, including (1) full description of our unified model in Section 4.2 and (2) more details about the annotation process in Section 3.3.

---

### Official Review · Reviewer_oV86 · 2021-07-23
**Interesting new task with dataset and metrics**

**Rating:** 8
**Confidence:** 3

**Review:**

Summary:
- The paper proposes the new task of hierarchical cross document coreference resolution (HCDCR), where mention clusters are organized in a hierarchy.  This can enable applications such as faceted query where the user can explore papers that refer to similar (but not necessarily the same) concepts (via a hierarchy).  To support this task, the authors contribute both a dataset (SciCO consisting of ~20K documents and ~26K mentions) and metrics for evaluation.  The dataset is constructed in a semi-automatic fashion, annotators are asked to annotate mention clusters and concept hierarchy, starting from candidate mention sets that are selected for annotators to label (by using a mix existing hierarchy of ML terms, automatically extracted hypernyms, and small curated list of lexically diverse concepts), while . Baselines (a coreference and a hierarchical model) and a unified model (predict both coreference and hierarchy) is proposed and compared on the dataset to provide an initial set of benchmark results.

Quality: The submission appears to be technically sound with well-thought out metrics and a reasonable set of experiments and natural initial models for the proposed task.

Clarity: The paper is very well written with clear description of the proposed task, metrics, data annotation process and experiments.

Originality: The paper contributes several original ideas (the HCDCR task, evaluation metrics, and proposed models - the models are mostly based on prior work and applied to the new task)

Significance: The work could stimulate research in a more fine-grained connections between documents.

Pros:
- The paper presents and interesting new task with dataset and metrics
- A initial model and set of experiments is provided

Cons:
- From a quick read over the paper, without careful examination of the data, it is difficult to judge the quality of the annotations and the dataset, and potential issues with the metrics.
- The automated construction of the dataset may introduce biases that favor algorithms that use the same resources/methods.
- It's unclear if the task would be of interest to the broader community or just a very specialized subset

---

> ### Author Response · Authors · 2021-07-28
> **Thanks for your review and feedback!**
>
> Thank you for your time reviewing our paper!
>
> We are encouraged that the reviewer finds that we contributed original ideas and that our tasks and metrics are interesting and sound.
>
> *RE quality of the dataset
>
> We would like to recall that the inter annotator agreement is relatively high according to both coreference and hierarchical links, indicating sufficient quality in SciCo. Notably, the IAA on coreference is even higher than the prominent Ontonotes corpus (as mentioned in Section 3.4), despite the specific challenges of the cross-document setting and the scientific domain. Additionally, we provide the full dataset in supplementary material for further examination of the annotation.
>
>
> *RE semi-automated construction of SciCo and bias
>
> We agree that the semi-automated construction of the dataset may introduce biases in SciCo. However, as discussed in the paper (Section 3.1), showing random mentions from a large corpus is ineffective as they will rarely be related. In fact, any non-random strategy for making the data collection feasible will introduce some form of bias in the dataset. In our work, we follow previous work on cross-document coreference annotation (e.g., Ravenscroft et al. “CD2CR: Co-reference Resolution Across Documents and Domains”; EACL 2021) and bootstrap the annotation. We use several resources (PwC, hypernym graph, and curated lists) in order to broaden the scope of the data, thus limiting possible biases.
>
>
> *RE: interest of broader community or specialized subset
>
> We believe our work could be useful for various research communities. The field of scientific information extraction has seen growing interest (Luan et al., EMNLP 2018, Hou et al, ACL 2019; Jain et al., ACL 2020; Hou et al., 2021), and coreference is an important and challenging subtask for IE, with no existing resources for cross-document coreference in science. SciCo also has the potential to enhance downstream applications at the corpus level, such as automated knowledge base construction (Hope et al., NAACL 2021; Mondal et al., NAACL 2021), citation recommendation (Bhagavatula et al., NAACL 2018), plagiarism detection (Zhou et al., EMNLP 2020), explaining relationships between documents (Luu et al., ACL 2021), etc.
>
> In addition, our task of hierarchical CDCR can serve as a general benchmark for several cross-document NLU challenges such as coreference and hierarchical relations based on multiple documents. As hierarchical relations are ubiquitous in language (e.g., whole-part, sub-events, etc.), our task could be explored in different settings from broader domains.

---

### Decision · Program_Chairs · 2021-08-17

**Decision:**

Accept

**Comment:**

This work introduces a new task of hierarchical cross-document coreference resolution, together with a new dataset and baselines for this task. Reviewers all agreed that this work makes strong contributions (the new task, evaluation metrics, and methods), and is well-written with clear description. Reviewers also raised minor concerns towards the quality of the dataset, annotation details, and the connection to prior work, for which the authors have provided detailed clarifications during the discussion stage. This is a clear accept and we encourage the authors to incorporate reviewers’ suggestions and their explanations in these authors' responses into the final version of the paper.